# Understanding and applying practitioner and patient views on the implementation of a novel automated Computer-Aided Risk Score (CARS) predicting the risk of death following emergency medical admission to hospital: qualitative study

Judith Dyson,[1] Claire Marsh,[2] Natalie Jackson,[2] Donald Richardson,[3] Muhammad Faisal,[4] Andrew J Scally,[5] Mohammed Mohammed[5]

For numbered affiliations see end of article.

**Correspondence to**
Dr Judith Dyson;
J.Dyson@hull.ac.uk

## ABSTRACT

**Objectives** The Computer-Aided Risk Score (CARS) estimates the risk of death following emergency admission to medical wards using routinely collected vital signs and blood test data. Our aim was to elicit the views of healthcare practitioners (staff) and service users and carers (SU/C) on (1) the potential value, unintended consequences and concerns associated with CARS and practitioner views on (2) the issues to consider before embedding CARS into routine practice.

**Setting** This study was conducted in two National Health Service (NHS) hospital trusts in the North of England. Both had in-house information technology (IT) development teams, mature IT infrastructure with electronic National Early Warning Score (NEWS) and were capable of integrating NEWS with blood test results. The study focused on emergency medical and elderly admissions units. There were 60 and 39 acute medical/elderly admissions beds at the two NHS hospital trusts.

**Participants** We conducted eight focus groups with 45 healthcare practitioners and two with 11 SU/Cs in two NHS acute hospitals.

**Results** Staff and SU/Cs recognised the potential of CARS but were clear that the score should not replace or undermine clinical judgments. Staff recognised that CARS could enhance clinical decision-making/judgments and aid communication with patients. They wanted to understand the components of CARS and be reassured about its accuracy but were concerned about the impact on intensive care and blood tests.

**Conclusion** Risk scores are widely used in healthcare, but their development and implementation do not usually involve input from practitioners and SU/Cs. We contributed to the development of CARS by eliciting views of staff and SU/Cs who provided important, often complex, insights to support the development and implementation of CARS to ensure successful implementation in routine clinical practice.

## Strengths and limitations of this study

► Our research takes a rare approach of including healthcare practitioner and service user/carer (SU/C) involvement in the development of a risk score.
► Finding adequate time for practitioner input was hard and we needed flexible approaches to focus group recruitment, venues and timings.
► Staff and SU/C input was largely a process of our consulting with the group. A codesign approach may have enhanced the benefits of stakeholder involvement.

## INTRODUCTION

A UK-wide study of 10 hospitals estimated 5% of deaths were preventable and 30% of these were attributable to poor clinical monitoring.[1] If risk of death information was available to clinical staff, it was likely to enhance patient safety[2]; however, there are no established risk equations for acutely admitted medical patients. Furthermore, while several studies have considered the use of physiological signs *or* blood tests in the assessment of patient, few consider combining the two.[2] Although National Early Warning Score (NEWS) is known to predict mortality in the hospital and prehospital setting,[3] it is not suitable for some groups of patients.[4] This research team therefore developed a novel Computer-Aided Risk Score (CARS) for estimating the risk of in-hospital mortality following emergency medical admission to hospital[5] in two hospitals. CARS was designed to rely on

variables already routinely collected and electronically recorded as part of the process of care including vital signs data (based on a NEWS)[6] and blood test results.[7] CARS demonstrated better discrimination and calibration than blood tests and NEWS separately.[5]

Despite the widespread use of risk scores to enhance decision-making in healthcare, as identified by Braband *et al*,[2] there is little or no documentary evidence of the involvement of healthcare practitioners and service users or carers (SU/Cs) in the design, development and implementation of this type of risk score. Our research responds to this gap.

Concurrent to the statistical modelling work, we conducted focus groups with healthcare practitioners and SU/Cs to feed into the ongoing development of CARS. The aims of the focus groups were to establish (1) healthcare practitioner (hereafter 'staff') and SU/C views on the potential value, unintended consequences and concerns associated with the development of the CARS and (2) staff views on how CARS should be adopted in practice/implementation needs.

## METHODS
### Patient and public involvement

All participants gave signed consent after receiving written information about the study. The 'Service User and Carer Involvement in Research Group' at the University of Bradford supported the project as members of the project steering group and as a focus group advisory team. Their contribution included codesign of project materials, support of the methodology (eg, recruitment strategies) and offering comments and suggestions based on data gathered.

### Participants

This study was conducted in two National Health Service (NHS) hospital trusts in the North of England (referred to hereafter as trust A and trust B). Both had in-house information technology (IT) development teams, mature IT infrastructure with electronic NEWS and were capable of integrating NEWS with blood test results. The study focused on emergency medical and elderly admissions units. There were 60 and 39 acute medical/elderly admissions beds at trusts A and B, respectively.

SU/Cs were competent adults (aged over 18 years) who were members of the public, who had either been in hospital themselves any time in their adult life, or who had experienced a relative in hospital. Staff were any practitioner working in areas where we intended to implement CARS (acute assessment units, medical wards and older person in-patient units) or acute outreach staff (nurse or doctor called on to offer advanced assessment and input) were eligible. Due to the additional aim of the staff groups (implementation needs), we held separate SU/C and staff focus groups.

### Design

Ten focus groups were held over two rounds in each trust (eight staff groups and two SU/C groups). A first round of staff and SU/C focus groups was conducted at the beginning of CARS development and commenced with a brief presentation about CARS, its rationale and development, then asked participants for their thoughts, feelings and concerns in relation to implementation of a CARS at their hospitals. Focus group schedules were informed by the literature relating to other risk scores (eg, NEWS). Results from focus groups were fed back to the CARS research team who then further developed CARS and its implementation package (figure 1 illustrates round one focus group questions). Subsequently, a second round of focus groups with staff explored CARS implementation needs in greater depth. After a presentation about CARS, vignettes were used (developed from case note reviews), to allow staff to understand how CARS scores relate to real clinical scenarios (figure 2 offers an example of a vignette and round two focus group questions). It was our intention to run groups of between 6 and 12 participants. Due to the challenges of staff time and availability, this was revised to smaller group sizes for the second staff focus groups. JD led staff groups, CM led SU/C groups and NJ supported all groups (none of whom had a previous relationship with participants and all were experienced in running focus groups).

### Procedure

CM approached the patient experience leads at each hospital as a gatekeeper for recruitment to SU/Cs. Patient experience leads contacted members of their forums through email, posters and verbal invites. Interested people contacted CM directly and were given participant information sheets before deciding to attend. Clinical partners from the CARS implementation teams at both hospitals introduced JD and NJ (electronically or in person) to the nurse in charge of relevant hospital areas. Charge nurses circulated email invitations to qualified medical and nursing staff of all grades with participant information. They contacted the research team if they were interested in participating. All interested SU/C and staff were included.

The first round of focus groups (SU/C and staff) took place over a 6-month period from May 2016 and the second staff focus groups occurred between May and July 2017. For staff, careful recruitment identified a diverse range of participants in terms of their professional role and experience. There were slightly more medical staff, which was expected given the relevance of the score. We expected focus groups including 25 participants within groups would be enough to achieve data saturation.[8] With one exception, all focus groups took place in hospital meeting rooms or offices. The exception was one staff focus group, which took place at a conference centre (for the convenience of staff).

### Analysis

Focus groups were audio recorded and transcribed verbatim and transcripts imported into NVivo V.11 data

| Staff focus group questions | Service User/Carer focus group questions |
|---|---|
| **General:** What are your thoughts on the CARS? How might the score be valuable? How might the score present challenges? Can you see any problems implementing the score in practice?<br>**Knowledge related to and presentation of the score:** What things do you still need to know about the CARS? What information do you think health care practitioners will need to use the score? How do you think this (information/score) is best disseminated to health care practitioners? How much weight might be attributed to the score in terms of clinical judgement? Talk us through how this might work. How might the score combine with clinical examination and patient reported symptoms?<br>**Components of the score:** Would you share your thoughts how accurate (reliable/valid) you consider the CARS to be based on the information/presentation we have given?<br>**Your responses to CARS:** Talk us through the likely process of action from receiving the CARS. How might the score might aid practitioners' decisions about treatment and care? Can you think of any circumstances where the score might support or undermine your clinical decision? Would you share the CARS score with patients? Relatives and carers? Are there any resource implications associated with the CARS?<br>**Organisational matters:** How might the CARS be used by the institution with regard to resource management? Is there any value to having score recorded in medical records? Any problems? | **General:** What are your thoughts and feelings about the CARS? How might the score be valuable? Do you have any worries about the score?<br>**Awareness:** What information about the score generally do you think people would like? What are your thoughts on patients being told about the score? What about relatives and carers?<br>**Impact on care:** The proposition is that the score might aid practitioners in treatment and care choices (e.g. admit/discharge home, where to admit, active treatment, supportive care)? How does this proposition sound to you? Does this proposition raise any concerns for you? How do you see the patient and carers role in this might be?<br>**Organisational matters:** How might the CARS be impact on resources/resource management or organisation of care?<br>**The following sentences were copied onto cards and presented in turn**<br>In response to the CARS practitioners may. . . . . . . . .<br>In response to the CARS I (as a patient)/patients may . . . . . . . . . .<br>In response to the CARS I (as SU/C)/my family . . . . . .<br>The CARS may be valuable for. . . . . . . . . .<br>The problems with the CARS might be. . . . . .<br>If I knew my/my relative's CARS I might feel . . . . . . . . . . .<br>The value for practitioners with regard to the CARS might be. . . . . . . . .<br>The problems for practitioners with regard to CARS might be. . . . . . . .<br>Practitioners should share the score with patients/carers when . . . . . . .<br>The best way to communicate the score might be. . . . . . .<br>Putting a number on my risk of deterioration is . . . . . . . |

**Figure 1** Focus group questions round 1. CARS, Computer-Aided Risk Score; SU/C, service user or carer.

management software. Data were subject to thematic analysis.[9] An inductive approach generated themes. Data were coded by one main coder (JD) but to increase reliability of analysis SU/C focus groups data were also coded by CM and a sample of staff focus group data coded by NJ. Coding was sentence by sentence to allow accurate comparison. Differences in coding were discussed and where necessary codes were redefined and the process repeated. On the second occasion, over 90% agreement was reached on codes allocated. Coding was according to the three areas of interest, value and unintended consequences, concerns and implementation. Data saturation was achieved; no new codes were derived from data from the last two focus groups.

| Vignette example | | Focus group questions |
|---|---|---|
| Patient Age (years): | [85] | **Response to Vignette:** Talk us through your decisions regarding the vignette (use up to 3 vignettes per group).<br>**Response to CARS:** The CARS score for this patient is [present score]. What are your thoughts on CARS given this context? Would the score change the decisions you made about treatment and care? How much would you attribute to the score in terms of clinical judgement? How might the score combine with clinical examination and patient reported symptoms?<br>**Knowledge related to the score:** How do you think this CARS is best implemented?<br>**Presentation of the score:** What did you think to the way we presented the score (scale of 1 to 8 with descriptors such as low, low, moderate and high)? How could it be better presented?<br>Finally, visual mock-ups of the score were presented and comments requested. |
| Gender: | [Male] | |
| time of admission: | [17:25] | |
| Mode of presentation: | [Acute/Emergency] | |
| **Description of presentation** | | |
| Recent previous admission with AKI and CKD, admitted as wife not coping – patient doubly incontinent with very poor mobility and new diarrhoea, past medical history of dementia, T2DM, previous stroke and osteoarthritis, on examination abdominal soft and non-tender, chest clear, heart sounds normal, commenced on IV fluids | | |
| Time of Index NEWS: | 00:50 NEWS [1] | |
| Respiratory Rate: | [18] breaths per minute | |
| Oxygen Saturations | [97] % | |
| Temperature | [35.1] degrees Celsius | |
| Systolic BP | [150] mmHg | |
| Diastolic BP | [65] mmHg | |
| Pulse Rate | [75] beats per minute | |
| Level of consciousness | [Alert] | |
| **Index blood test results** | | |

| Blood Test | Result |
|---|---|
| Albumin (g/L) | [35] |
| Creatinine (umol/L) | [986] |
| Haemoglobin (g/dl) | [103] |
| Potassium (mmol/L) | [5.2] |
| Sodium (mmol/L) | [142] |
| Urea (mmol/L) | [38.2] |
| White blood cell count ($10^9$ cells/L) | [5.5] |
| Platelets ($10^3$/microliter) | [281] |
| Acute Kidney Injury Score | [3] |

**Figure 2** Focus group content round 2. AKI, acute kidney injury; BP, blood pressure; CARS, Computer-Aided Risk Score; CKD, chronic kidney disease; IV, intravenous; NEWS, National Early Warning Score; T2DM, type 2 diabetes mellitus.

| Practitioner | Round 1 | | Round 2 | | | | | | Total |
|---|---|---|---|---|---|---|---|---|---|
| | A[1] | B[2] | A[3] | A[4] | A[5] | B[6] | B[7] | B[8] | |
| | n=7 | n=10 | n=16 | n=2 | n=3 | n=3 | n=2 | n=2 | n=45 |
| Doctor (Dr) | 5 | 2 | 6 | 0 | 3 | 1 | 0 | 0 | 17 |
| (Senior (Sr) doctor (Dr) consultant/senior registrar) | (3) | (2) | (6) | (0) | (0) | (1) | (0) | (0) | (12) |
| (Junior doctor (Jr) registrar, FY2, FY1) | (2) | (0) | (0) | (0) | (3) | (0) | (0) | (0) | (5) |
| Ward based Nurse (N) | 0 | 2 | 6 | 2 | 0 | 1 | 2 | 2 | 15 |
| Senior (Sr) Nurse (N) (above band 6) | (0) | (2) | (5) | (2) | (0) | (0) | (2) | (0) | (11) |
| Junior (Jr) Nurse (N) (below band 6) | (0) | (0) | (1) | (0) | (0) | (1) | (0) | (2) | (4) |
| Nurse Specialist (NS) | 2 | 5 | 0 | 0 | 0 | 0 | 0 | 0 | 7 |
| Health Care Assistant (HCA) | 0 | 0 | 2 | 0 | 0 | 0 | 0 | 0 | 2 |
| Other (O) (allied professionals) | 0 | 1 | 2 | 0 | 0 | 1 | 0 | 0 | 4 |

**Figure 3** Staff focus group participants.

## RESULTS

### Characteristics of the sample

SU/C groups in NHS trusts A and B involved six and five participants, respectively. The composition of the staff groups was according to figure 3; junior doctor refers to doctors in their first (FY1) or second (FY2) year post qualification or registrar (first promotion post qualifying). Senior doctor refers to senior registrar (preconsultant grade) and consultant (most senior medical person). Grades of nurses include below six (five being the most junior post qualification nurse) and above six (seven charge nurse, eight matron and above nurse specialists or clinical managers). Allied professionals included physiotherapists and occupational therapists. We did not formally ask SU/Cs about the nature of their or their friend/relative's hospital stay; however, examples given to support views offered suggests a wide range of experiences (eg, heart bypass surgery, caring at the end of life, chest infection and subsequent pneumonia). SU/Cs talked about admissions to the emergency department, intensive care unit and both medical and surgical wards. There were eight staff focus groups with the number of participants ranging from 2 to 16 across both trusts. There were six and five participants in SU/C groups in trusts A and B, respectively. All participants contributed to focus groups. The duration of focus groups ranged between 22 min and 1 hour 29 min, mean duration 57 min.

### Overall findings

There were nine themes arranged according to the aims of the study, 'value and unintended consequences', 'concerns' and 'implementation' represented in figure 4 and elaborated below with verbatim quotes from a broad range of participants.

### Value and unintended consequences

#### Decision-making and clinical judgement

Staff talked about the value of using CARS as a decision aid for choosing active or supportive care or for 'do not resuscitate' decisions.

> …on admission… might help triage Sr Dr1 FG1

> …decisions about end of life care as well; guide DNR [do not resuscitate] decision making Sr Dr3, FG1

This was considered within different *contexts* of care; in some areas a high score might suggest supportive care (eg, general medical areas) and in other areas (eg, paediatrics) it would be expected even the smallest chance of survival suggests active care.

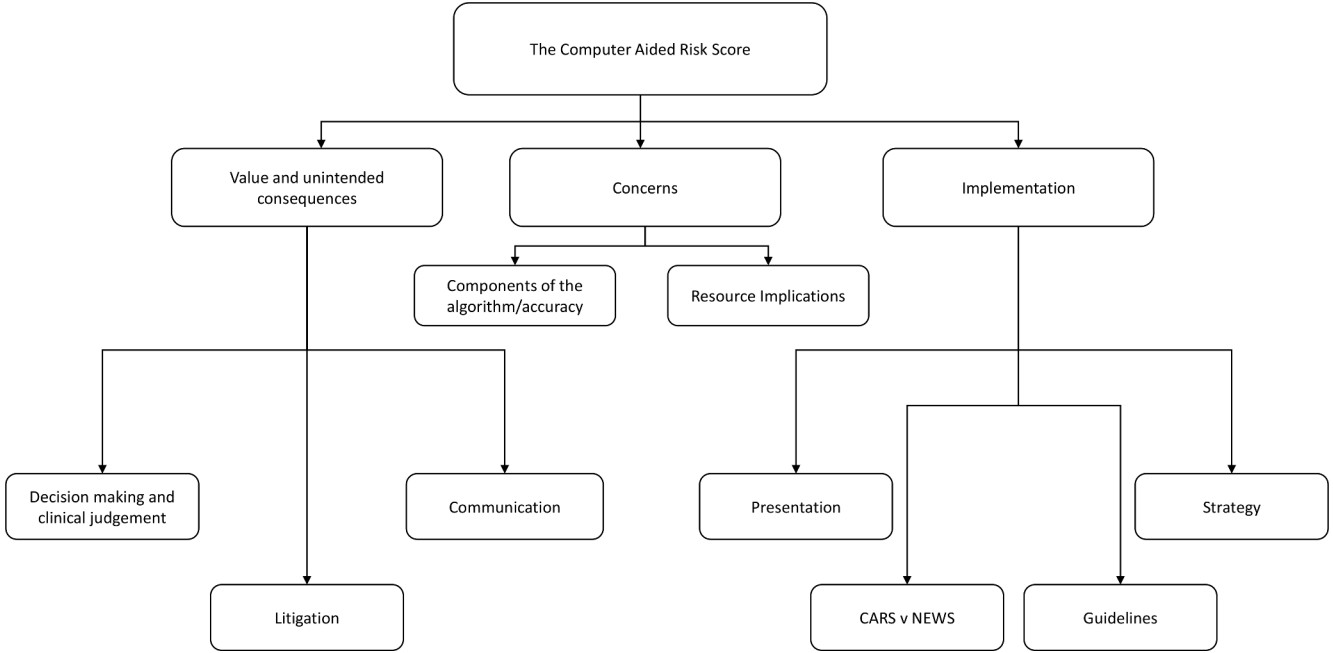

**Figure 4** Themes resulting from data analysis according to the study aims. CARS, Computer-Aided Risk Score; NEWS, National Early Warning Score.

some clinicians would say even half a percent chance of survival is enough… where do we draw the line? Sr Dr3, FG1

Staff considered the score would give them extra confidence in making clinical decisions.

Having the CARS then makes you think no ok I am on the right way and it gives you a bit more confidence Sr N1, FG3

I think CARS score, well all scoring systems are useful anyway as they are a starting point to see how sick your patient is, it's a good way of just getting all the information in one go Sr Dr1, FG3

SU/C groups were similarly positive about its potential to play a role in improving patient safety because it could challenge clinician's preconceived ideas and provide additional information for clinicians to make a judgement.

what I've seen is that they get this idea in their head of what's wrong with you and this is perhaps a good way of making double check SU/C Trust B

Both staff and SU/Cs expressed concerns that clinical judgement may be undermined by the score. Staff discussions focused on the appropriate 'weight' to give the score in the decision-making process particularly when the score and their own judgement conflicted.

We want the space to use that judgement… Jr Dr1 FG1

With all of these things [scores] you stop taking a clinical interest in the patient and just look at the numbers NS1 FG2

as long as it's another helpful factor in deciding what to do as opposed to being the determining factor SU/C Trust A

## Litigation

Staff saw potential positive and negative elements to the score in terms of supporting their decisions or otherwise and the potential for litigation.

If you need to back up your clinical judgement to the coroner [the Computer Aided Risk Score] would help I think Sr Dr2 FG1

Someone is going to say, I am going to pore over those notes and find out why this person has died, whereas, they may not have done previously. Conversely this person had a high chance of dying, why did you carry on with your treatment which… was futile Sr Dr1 FG1

## Communication

Most staff considered the score would aid or prompt communication with patients about prognosis. There was disagreement about it might be best to give the actual score or a description of the situation.

We tend to use more descriptive terms… patients are appreciative of honesty Sr Dr1 FG1

If he had a score, today this is how bad she actually is its likely going to be soon, that would have helped him deal with the situation better SU/C Trust A

SU/C participants talked at length about the score's potential role in assisting communication about a patient's condition, helping them accept the seriousness of the condition. However, there was much debate among both staff and SU/Cs about whether or not patients and carers should be actively informed of the score.

One of the biggest things in any hospital anywhere is communication and information they're not told, and I'm sure they would like to be told SU/C Trust B

They [staff] are a little too tactful, little too polite, little too sensitive. . the score might help' SU/C Trust B

It was argued this score was primarily for clinicians and while it should not hidden, it need not be routinely provided in the same format for everyone.

I think if the family are told they are gravely ill that would be more human than they are an eight point four SR Dr1 FG2

## Components of the algorithm/accuracy

Staff discussed at length the need to know the component parts of the score and access the latest contributing values. They were keen to know when CARS may or may not be accurate. Frequent examples offered of where CARS may not be accurate were chronic obstructive pulmonary disease, chronic kidney disease terminal conditions and congenital diseases.

I would like the people who are reviewing the score to be able to understand it properly. Otherwise you will get people who are becoming overly worried about it when they don't actually, can't interpret it and don't understand it. Jr Dr1 FG5

What about COPD [Chronic Obstructive Pulmonary Disease]… at the same time people with significant co-morbidities will have a worse outcome? Jr Dr2 FG1

SU/C groups were also interested in exploring the accuracy of the score:

It needs to be sensitive both ways… otherwise everybody will be in the high dependency unit SU/C Trust B

## Resource implications

Staff raised questions about the potential resource needs associated with CARS, for example, intensive care unit beds and blood tests.

Can I just ask what sort of impact this will have on the labs? NS2 FG2

the extra expenditure… you have an ethical dilemma because you have a patient who's got a score you've gone to escalating to HDU [High Dependency Unit],

ICU [Intensive Care Unit], high observations units. NS3 FG2

SU/C groups were also concerned about resources but this focused on extra workload and they were worried the score would mean less face-to-face care.

My concerns would be they're already under extreme pressure, if this is going to be another assessment that they have got to carry out on patients that's increasing the pressure at a time when they're already stressed out SU/C Trust B

### Implementation
#### Presentation
Staff focus groups had ideas as to the presentation of CARS and the need to see the trend in the score.

If I were using it myself as a physician I would want a specific percentage Jr Dr1 FG1

Putting it all together in a score is helpful when you are the person on call who doesn't know them and you can see the trend of that score and it's helpful. Jr Dr1 FG5

It would be useful to see it as a graph [trend] Jr Ns1 FG8

When discussing presentation of the score, the SU/Cs focused mainly on the communication of it by staff (reported above) and whether they should have direct access/sight of the score.

To see it change in front of your eyes that might be even more terrifying SU/C Trust A

#### CARS compared with NEWS
Staff appreciated that CARS was more sensitive than NEWS, and though they appreciated, unlike NEWS, the CARS was a complex statistical equation, not possible to calculate by hand, they were keen to see it.

Can you copy that algorithm? Can we have a look at that? NS3 FG2 Trust B

The key issue with respect to this comparison was the potential for CARS to suggest one action should be taken and NEWS suggests another (eg, one indicates escalation the other does not).

NEWS [National Early Warning Score] score high and CARS low or vice versa therefore we've then got a confusion to the people who are actually on the shop floor where one thing is telling them to do this escalation and the other is saying you don't need to O FG2

Staff considered the comparative utility of NEWS and CARS with a particular focus on whether blood tests would delay a calculation of the score, or, whether CARS would be updated when any new data item (eg, temperature/pulse) became available.

[CARS unlike NEWS] *it might take three or 4 hours… if it relies on blood tests* Sr Dr3 FG1

Finally, there was indication staff wanted to see the score demonstrated to be effective in relation to *people* in addition to being mathematically valid.

#### Guidelines
Staff discussed the specific procedures for CARS' role in confirming or support clinical judgement:

I don't think you can use the CARS score as a trigger to make any specific clinical action, it is an alert that there may be a clinical problem there, there is a clinical problem there and then you need to find out what it actually is, it may be you need to look more closely at the biochemistry or whatever, whereas, the NEWS score is more specific NS4 FG2

They initially suggested the need for an escalation protocol or guide (where actions are prescribed according to score).

…with the NEWS [National Early Warning Score] if you have a score of five an above obviously that is an escalation process whereas with the CARS we don't know NS1 FG2

When vignettes were brought in during the second staff focus groups, staff were less likely to feel the need for an escalation protocol or a guide.

I don't think it [CARS] changes the clinical management because the clinical management is always going to be based on the individual in front of you with their individual bloods and things. Putting it all together in a score is helpful Jr Dr2 FG5

Where staff wanted guidance, this was sometimes to protect against *criticism* about inappropriate response to a high CARS.

If there isn't [a protoco], you go to the doctor and say the CARS has come up at this score, and they say yes that's because. . . . . they would always have a rationale… but at least you are covering yourself Sr Ns1 FG7

This was linked to concerns about litigation.

Then one day someone will turn around and say but the CARS score was 10 and you didn't do this, so I think that's just the world we live in and we have to have a clear role when we introduce it or otherwise… Jr1 Dr FG 5

Some felt the guidance would ensure (insist on) a response from a senior clinician.

You can say to the doctor look I am just following the protocol Sr Ns1 FG4

Others suggested that it would serve as support for more junior staff.

Junior medical and nursing staff . . . I would worry that they would, that they might lack clinical prowess Sr Dr3 FG3

### Strategy

There was a lot of discussion about a strategy for implementation. One group (FG2) spent a lot of time discussing the need for a 'champion'. This related to their experience with NEWS. The suggestion was one of the outreach team would be best placed for this. Discussion took place about the extent of education required and there was a consensus that a hospital-wide strategy would be appropriate.

We need people to champion this O FG2

The big nooks and crooks is going to be education, needs training, information as well. If you're just looking at one area, there are medical people coming through that area teams to you need to target them all. Sr Dr1 FG2

### DISCUSSION

Our approach in developing CARS has been coworking with front-line staff and SU/Cs as part of the project team as well as participants. We conducted 10 focus groups with 11 SU/Cs and 45 healthcare practitioners in two NHS acute hospitals. Participants were interested in the development of the CARS score and appreciated such efforts to improve patient safety at their hospital. They recognised the potential of CARS but were clear that the score should not replace or undermine clinical judgments. Staff recognised that CARS could enhance clinical decision-making and aid communication. They wanted to understand the components of CARS and be reassured about its accuracy and were concerned about the impact on resources. Staff preferred CARS to be shown as a score (without descriptive labels) graphed by time to monitor changes. Staff needed clarity on how CARS and NEWS would work alongside each other. SU/C has mixed views about the extent to the score should be shared with patients.

As far as we are aware, previous studies on the design, development and implementation of risk scores have not reported on the views of staff and SU/C, so we are unable to determine the extent implementation of these risk-scoring systems into routine clinical practice that requires careful consideration of the views of staff and patients. However, our broad recruitment and data saturation suggests that our findings may be generalisable to the implementation of risk scores elsewhere. The themes identified highlight that risk scores are complex interventions introduced into complex adaptive systems and the voice of staff and SU/C is an important element of codesign. The contribution of staff and SU/C was integral and iterative to the design of CARS and led to some important insights and design changes including: (1) CARS will update over time and be available as a graph with all its subcomponents also shown. (2) The relationship of CARS with NEWS was important to clarify. We have now designed our risk score to use NEWS in the first instance and then incorporate blood test results as and when available. About one-fourth of patients do not have a blood test results. (3) We have decided to present the score as a % (0–100) without descriptive labels (eg, low/medium/high). (4) The score will be visible on the electronic patient record but will not be a 'pop-up' alert. We are now working with both NHS trusts, taking a staged approach to implementing CARS as a quality improvement programme and we continue to involve staff and SU/Cs. The qualitative work reported here continues to map the process, identify early problems and support solutions.

The process of involvement of stakeholders within interventions is challenging and we can usefully reflect on the limitations of our efforts. For staff, finding time for their quality input was hard and while we achieved this through flexible approaches to recruitment, venues and timings, the process revealed warnings about the 'unfinished business' and 'unanswered questions' that staff still have about implementation. With respect to SU/Cs, we can refer to notions of an involvement hierarchy of 'consultation, collaboration, and user-control'[10] to critique our approach. We predominantly 'consulted'; however, we extended this to 'collaboration' with our steering group members who maintained input throughout. Much of the implementation and research took place at individual sites, integrated into daily hospital working, mirroring iterative quality improvement process. Including SU/Cs in this is fraught with difficulty and is currently rare in healthcare.[11] Nevertheless as healthcare is increasingly using computer-aided decision support systems as a key to achieving gains in quality and patient safety,[12] we suggest that codesign is necessary to maximise the successful implementation.

### Conclusions

Staff and patients had important, often complex, insights to support the development and implementation of CARS which need to be addressed if CARS is to be successfully used in routine practice.

**Author affiliations**
[1]Health and Social Work, University of Hull, Hull, East Riding of Yorkshire, UK
[2]Quality and Safety, Bradord Institute for Health Research, Bradford, UK
[3]Renal Medicine, York Teaching Hospital NHS Foundation Trust Hospital, York, UK
[4]Faculty of Health Studies, University of Bradford, Bradford, West Yorkshire, UK
[5]School of Health Studies, University of Bradford, Bradford, UK

**Contributors** MM and DR had the original idea for this work. NJ was the study coordinator. JD, CM and NJ were the leads for the qualitative study. JD, NJ, CM and MM wrote the first draft of this paper. All authors (including AJS and MF) subsequently assisted in redrafting and have approved the final version.

**Funding** This study is funded by the Health Foundation and the National Institute for Health Research (NIHR) Yorkshire and Humber Patient Safety Translational Research Centre (NIHR Yorkshire and Humber PSTRC). The Health Foundation is an independent charity working to improve the quality of health care in the UK.

**Disclaimer**  The views expressed in this article are those of the author(s) and not necessarily those of the Health Foundation, the NHS, the NIHR, or the Department of Health and Social Care.

**Competing interests**  None declared.

**Patient consent for publication**  Next of kin consent obtained.

**Ethics approval**  Ethical approval was granted from National Research Ethics Service (15/YH/0348) and Research Governance approval from the trusts involved in the study.

**Provenance and peer review**  Not commissioned; externally peer reviewed.

**Data sharing statement**  No additional data are available.

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
