## [Reviewer comments · BMJ Open]

ARTICLE DETAILS

TITLE (PROVISIONAL)	Understanding and applying practitioner and patient views on the implementation of a novel automated Computer Aided Risk Score (CARS) predicting the risk of death following emergency medical admission to hospital: A qualitative study
AUTHORS	Dyson, Judith; Marsh, Claire; Jackson, Natalie; Richardson, Donald; Faisal, Muhammad; Scally, Andrew; Mohammed, Mohammed

VERSION 1 - REVIEW

REVIEWER	Rebecca Randell University of Leeds, UK
REVIEW RETURNED	07-Dec-2018

GENERAL COMMENTS	This is an interesting paper, providing implications for the design of computer-based risk scores, and generally well written. However, currently the authors are not making the most of the data that they have collected - the analysis seems underdeveloped. A key issue is that effort was made to include a range of healthcare professional groups in the focus groups yet the analysis does not indicate similarities/differences in perspectives of these different professional groups. Please make clear whether particular perspectives were voiced predominantly by one professional group or whether it was a concern that appeared to be shared across professional groups. The presentation of the findings could also be improved - I would advise combining some of the sub-sections because the very short sub-sections break up the flow of the text. Specifically, I would combine decision making and clinical judgement - these two processes are intimately linked so I don't see it as helpful to separate them out. I would also argue that 'components of the algorithm' and 'accuracy' are also elements of judgement and decision making. At a couple of points in the findings there appears to be a conflict between the narrative and the quotes: - At the bottom of page 8, it's said that there was disagreement about whether or not the actual score should be given but a quote representing only one of these perspectives is given. A quote representing the alternative perspective isn't given until the end of that sub-section (following discussion of service user/carer perspectives) - this should be moved up.- Under components, it is stated that 'Staff discussed at length...' yet only one quote is given and no further information about that discussion is provided. The following changes are also needed:
---

	 - In the introduction, please give more information about the rationale for the CARS - what were your expectations for how it would inform patient care? - In the methods, please clearly describe the breakdown of focus groups between rounds and between staff and service users/carers - on page 4 and in the abstract it says 8 focus groups were undertaken, but then its later stated that 8 staff focus groups were undertaken. Please also clearly state how many participants were in each focus group (this is done for the staff focus groups via Figure 3 but not for service users/carers). - Please state whether or not those who led the focus groups had experience of doing so. - When you say (page 5) that focus groups including 25 participants was expected to receive data saturation, I'm assuming you mean 25 over multiple focus groups, not 25 within a focus group - please rephrase the statement to make this clearer. - Page 6, l.7 - please explain how you determined that data saturation had been achieved. - Page 6, l.35 - please state the average duration of the focus groups. - In the discussion, please reflect on the benefits and limitations of focus groups. Did certain people dominate the discussion? For example, medical staff are far more represented in the quotes than nursing staff - is that a reflection of the nature of the discussion? Please also, with your quotes, number participants individually (e.g. Sr Dr 1, FG1 rather than just Sr Dr, FG) so that readers can judge for themselves whether quotes are predominantly from particular individuals. - Finally, I did notice some typos (e.g. p.3, l.39 should be 'development of a', p.4, l.55 'with were fed') so please carefully proof-read.
--	---

REVIEWER	Alison Porter Swansea University
REVIEW RETURNED	27-Dec-2018

GENERAL COMMENTS	This paper is generally well written and interested, but could do with attention at a few points. Title does not seem to accurately reflect the objectives - it focuses on participation in implementation, whereas the objectives are about views/attitudes. The introduction is very short and under-referenced - there is much more you could say about risk scores, and also about implementation processes. Methods - how were the service users/carers recruited and selected? Did staff and SU/C feed into the design of the CARS? it's not entirely clear. Results - characteristics of sample: how many service users, how many carers? Discussion section needs to make more reference to previous literature. Currently the reference list is very short and only includes one work on risk scores - written by the authors of this paper.
--

VERSION 1 – AUTHOR RESPONSE

Reviewer: 1 Rebecca Randell

4. This is an interesting paper, providing implications for the design of computer-based risk scores, and generally well written.

Thank you.

5. However, currently the authors are not making the most of the data that they have collected - the analysis seems underdeveloped. A key issue is that effort was made to include a range of healthcare professional groups in the focus groups yet the analysis does not indicate similarities/differences in perspectives of these different professional groups. Please make clear whether particular perspectives were voiced predominantly by one professional group or whether it was a concern that appeared to be shared across professional groups.

Thank you for this comment. We agree we did not separate SU/C and staff findings in this paper. We have added to the second paragraph in the discussion to clarify the similarities and differences between the groups (which were few).

6. The presentation of the findings could also be improved - I would advise combining some of the sub-sections because the very short sub-sections break up the flow of the text. Specifically, I would combine decision making and clinical judgement - these two processes are intimately linked so I don't see it as helpful to separate them out. I would also argue that 'components of the algorithm' and 'accuracy' are also elements of judgement and decision making.

We are happy with this suggestion and have made the suggested changes which can be seen in figure 4 and on pages 6, 7 and 9.

7. At a couple of points in the findings there appears to be a conflict between the narrative and the quotes: At the bottom of page 8, it's said that there was disagreement about whether or not the actual score should be given but a quote representing only one of these perspectives is given. A quote representing the alternative perspective isn't given until the end of that sub-section (following discussion of service user/carer perspectives) - this should be moved up.

This seems sensible and we have made this change on page 8.

8. Under components, it is stated that 'Staff discussed at length...' yet only one quote is given and no further information about that discussion is provided.

By combining themes as suggested in point 6 we have addressed this point. There are now several quotes supporting this statement.

9. The following changes are also needed: In the introduction, please give more information about the rationale for the CARS - what were your expectations for how it would inform patient care?

We have added further information in paragraph 1 on page 3.

10. In the methods, please clearly describe the breakdown of focus groups between rounds and between staff and service users/carers - on page 4 and in the abstract it says 8 focus groups were undertaken, but then it's later stated that 8 staff focus groups were undertaken. Please also clearly state how many participants were in each focus group (this is done for the staff focus groups via Figure 3 but not for service users/carers).

We have added this extra detail in the abstract on page 1 and in the section "design" on page 4. We have also stated the number of service users in each group.

11. Please state whether or not those who led the focus groups had experience of doing so.

We have added this detail on page 5.

12. When you say (page 5) that focus groups including 25 participants was expected to receive data saturation, I'm assuming you mean 25 over multiple focus groups, not 25 within a focus group - please rephrase the statement to make this clearer.

Yes this is correct; we have clarified participants within groups rather than groups.

13. Page 6, l.7 - please explain how you determined that data saturation had been achieved.

We have added this detail to the sentence indicated.

14. Page 6, l.35 - please state the average duration of the focus groups.

We have added the mean duration of focus groups on the page and line indicated.

15. In the discussion, please reflect on the benefits and limitations of focus groups. Did certain people dominate the discussion? For example, medical staff are far more represented in the quotes than nursing staff - is that a reflection of the nature of the discussion? Please also, with your quotes, number participants individually (e.g. Sr Dr 1, FG1 rather than just Sr Dr, FG) so that readers can judge for themselves whether quotes are predominantly from particular individuals.

We have indicated individual participants by using a number as suggested here. We have added a sentence indicating that the score is more relevant to medical compared with other staff (pg 5) and this may account for a disproportionate number of medical participants however, all participants contributed to the focus group discussions (end of page 6).

16. Finally, I did notice some typos (e.g. p.3, l.39 should be 'development of a', p.4, l.55 'with were fed') so please carefully proof-read.

Thank you for picking these up; we have made these specific changes and carefully proof re-read the paper.

Reviewer: 2 Alison Porter

17. This paper is generally well written and interested, but could do with attention at a few points.

Thank you.

18. Title does not seem to accurately reflect the objectives - it focuses on participation in implementation, whereas the objectives are about views/attitudes.

We have tweaked the title to reflect "understanding and applying practitioner and patient views".

19. The introduction is very short and under-referenced - there is much more you could say about risk scores, and also about implementation processes.

We have added more narrative and references according to point 9 above (reviewer 1).

20. Methods - how were the service users/carers recruited and selected?

We have added extra words in the "procedure" section on page 5 to clarify recruitment and selection.

21. Did staff and SU/C feed into the design of the CARS? it's not entirely clear.

Yes, we have clarified this in the second paragraph of our discussion.

22. Results - characteristics of sample: how many service users, how many carers?

We would prefer not to distinguish this as we didn't define the participants to this level of detail. A number of members of the group had been both carers and service users.

23. Discussion section needs to make more reference to previous literature. Currently the reference list is very short and only includes one work on risk scores - written by the authors of this paper.

We added more references relating to risk score development in the introduction. This illustrates the gap in the literature relating to co-design and thus justifies fewer references in the discussion.

VERSION 2 – REVIEW

REVIEWER	Rebecca Randell University of Leeds, UK
REVIEW RETURNED	13-Feb-2019

GENERAL COMMENTS	The authors have adequately addressed my comments and I recommend the manuscript for publication.
---